# ERK-Directed Phosphorylation of mGlu5 Gates Methamphetamine Reward and Reinforcement in Mouse

**DOI:** 10.3390/ijms22031473

**Published:** 2021-02-02

**Authors:** Elissa K. Fultz, Sema G. Quadir, Douglas Martin, Daniel M. Flaherty, Paul F. Worley, Tod E. Kippin, Karen K. Szumlinski

**Affiliations:** 1Department of Psychological and Brain Sciences, University of California Santa Barbara, Santa Barbara, CA 93106, USA; elissa.fultz@gmail.com (E.K.F.); semaquadir@gmail.com (S.G.Q.); douglas@martinhome.net (D.M.); dannyflare99@gmail.com (D.M.F.); kippin@ucsb.edu (T.E.K.); 2Department of Neuroscience, Johns Hopkins University School of Medicine, Baltimore, MD 21205, USA; pworley1@jhmi.edu; 3Department of Molecular, Cellular and Developmental Biology and the Neuroscience Research Institute, University of California Santa Barbara, Santa Barbara, CA 93106, USA; 4Institute for Collaborative Biotechnologies, University of California Santa Barbara, Santa Barbara, CA 93106, USA

**Keywords:** metabotropic glutamate receptor 5, place-conditioning, reinforcement, self-administration, vulnerability, methamphetamine, addiction

## Abstract

Methamphetamine (MA) is a highly addictive psychomotor stimulant drug. In recent years, MA use has increased exponentially on a global scale, with the number of MA-involved deaths reaching epidemic proportions. There is no approved pharmacotherapy for treating MA use disorder, and we know relatively little regarding the neurobiological determinants of vulnerability to this disease. Extracellular signal-regulated kinase (ERK) is an important signaling molecule implicated in the long-lasting neuroadaptations purported to underlie the development of substance use disorders, but the role for this kinase in the propensity to develop addiction, particularly MA use disorder, is uncharacterized. In a previous MA-induced place-conditioning study of C57BL/6J mice, we characterized mice as MA-preferring, -neutral, or -avoiding and collected tissue from the medial prefrontal cortex (mPFC). Using immunoblotting, we determined that elevated phosphorylated ERK expression within the medial prefrontal cortex (mPFC) is a biochemical correlate of the affective valence of MA in a population of C57BL/6J mice. We confirmed the functional relevance for mPFC ERK activation for MA-induced place-preference via site-directed infusion of the MEK inhibitor U0126. By contrast, ERK inhibition did not have any effect upon MA-induced locomotion or its sensitization upon repeated MA treatment. Through studies of transgenic mice with alanine point mutations on T1123/S1126 of mGlu5 that disrupt ERK-dependent phosphorylation of the receptor, we discovered that ERK-dependent mGlu5 phosphorylation normally suppresses MA-induced conditioned place-preference (MA-CPP), but is necessary for this drug’s reinforcing properties. If relevant to humans, the present results implicate individual differences in the capacity of MA-associated cues/contexts to hyper-activate ERK signaling within mPFC in MA Use Disorder vulnerability and pose mGlu5 as one ERK-directed target contributing to the propensity to seek out and take MA.

## 1. Introduction

In the wake of the opioid epidemic, the world is currently facing a psychostimulant crisis. The Drug Enforcement Agency of the United States reports a recent, major surge in the amount of methamphetamine (MA) seized by law enforcement, with a 15-fold higher supply and demand for psychostimulants than opioids world-wide [1]. By 2017, the total number of U.S. persons reporting past year stimulant use is higher than that for opioids [1], with a similar trend observed in other countries, particularly in Asia [1,2]. Accordingly, the Center for Disease Control of the United States and the World Health Organization report that deaths involving MA have risen rapidly to reach epidemic proportions [1,2]. However, unlike Opioid Use Disorder, no clinically approved pharmacotherapy exists for treating Methamphetamine Use Disorder. 

MA is an indirect dopamine agonist [3,4] and profoundly increases DA concentrations within forebrain terminals, in particular the nucleus accumbens, dorsal striatum, and prefrontal cortex (PFC) [5]. Indeed, the bulk of neurobiological research regarding Methamphetamine Use Disorder in humans has focused upon forebrain dopamine systems, e.g., [6,7,8,9], with a good majority of the extant basic science studies in rodent models of Methamphetamine Use Disorder employing very high-dose MA treatment regimens that elicit dopamine neurotoxicity selectively within dorsal striatal regions, cf. [10,11,12]. While such an approach provided tremendous biochemical insight into how high-dose MA induces forebrain damage of relevance to late-stage addiction, the question of why only certain individuals come to repeatedly abuse MA in the first place and an understanding of the neurobiological substrates underpinning risk for MA Use Disorder have received relatively less experimental attention, cf. [13].

In humans, moderate doses of amphetamine-type stimulants (e.g., 0.1–0.4 mg/kg) elicit euphoria and behavioral activation, which are typically considered appetitive/reinforcing. Higher, subtoxic, MA doses (e.g., 1.0–4.0 mg/kg) can induce anxiety, cause severe headaches, and stimulate the cardiovascular system, which can be perceived as aversive, cf. [14,15]. As for other drugs of abuse, e.g., [16,17,18,19,20,21], individual differences in sensitivity to MA’s rewarding/aversive effects likely influence risk of continued MA abuse/addiction in humans, which is corroborated by prior literature using murine models of MA Use Disorder risk/resiliency [22,23,24,25,26]; cf. [13]. Of relevance to this report, there is accumulating evidence implicating corticoaccumbens glutamate transmission in gating the affective valence of MA and subsequent drug-taking behavior [26,27,28,29], with additional evidence supporting a key role for the mGlu5 subtype of glutamate receptor in MA preference, MA self-administration, and reactivity to MA-associated cues [30,31,32,33,34,35,36]. While reports of the expression levels or functional status of mGlu5 in the brains of humans with MA Use Disorder are forthcoming (https://grantome.com/grant/NIH/R33-DA031441-06), MA Use Disorder is associated with lower levels of glutamatergic neurometabolites in frontal cortical regions [37,38,39], supporting a key role for corticoaccumbens glutamate dysfunction in MA Use Disorder vulnerability and severity. 

Extracellular signal-related kinase (ERK) is a member of the mitogen-activated protein kinase (MAPK) signaling cascade and has long been implicated in the neurobiology of Substance Use Disorder, cf. [40,41,42,43]. However, its role in MA Use Disorder vulnerability is under-studied, with animal research focused on the nucleus accumbens [44,45]. We reported previously that blunted MA-induced glutamate release and ERK activity within the ventromedial aspect of the PFC (vmPFC) were biochemical traits correlated with selection for high versus low MA-drinking in mice [27]. Here, we extend these immunoblotting findings to isogenic mice spontaneously exhibiting high MA conditioned place-preference (MA-CPP) versus -aversion and examine the functional consequences of pharmacologically targeting ERK activation within vmPFC for MA-conditioned reward/aversion. ERK phosphorylates mGlu5 on T1123 and S1126 and mice with alanine substitutions at these sites exhibit altered behavioral responses to both cocaine [46] and alcohol [47]. Thus, we also characterized MA reward and reinforcement in mGlu5^T1123A/S1126A^ mice (*Grm5^AA/AA^*) and their wild-type (WT) controls (*Grm5^TS/TS^*) to probe mGlu5 as a potential site of ERK-dependent regulation of MA reward and reinforcement. 

## 2. Results

### 2.1. pERK Expression within mPFC Predicts the Positive Affective Valence of MA

The CPP Scores (following conditioning with 2 mg/kg, IP) obtained from male B6J mice categorized as CPP, Neutral, and CPA are provided in Figure 1A (see Ref. [26] for detailed analyses of the place-conditioning behavior of these mice). Immunoblots of the mPFC failed to detect any group differences in total ERK expression within mPFC (Figure 1C, right; univariate ANOVA, *p* = 0.17). In contrast, an examination of total and the relative expression of pERK indicated higher ERK activity within the mPFC of CPP animals, relative to the other groups tested (Figure 1C). This observation was confirmed by ANOVA (for total pERK, F(1,49) = 5.53, *p* = 0.003; LSD post-hoc tests, CPP vs. SAL: *p* = 0.003; CPP vs. Neutral: *p* = 0.006; CPP vs. CPA: *p* = 0.001; for ratio: F(1,49) = 4.71, *p* = 0.006; LSD post-hoc tests, CPP vs. SAL: *p* = 0.002; CPP vs. Neutral: *p* = 0.006; CPP vs. CPA: *p* = 0.007). Further, both total pERK expression (r = 0.056, *p* < 0.005, N = 38; data not shown) and the ratio of pERK:ERK within mPFC predicted the magnitude of the place-conditioned response (Figure 1D; r = 0.69, *p* < 0.0001, N = 38). Thus, MA-preference is predicted by the activational state of ERK within mPFC of isogenic mice.

### 2.2. U0126 Reduces MA-Induced Place-Preference

We next sought to gain causal evidence of a role for ERK in MA-preference under place-conditioning procedures in a group of B6J male mice distinct from those employed in the immunoblotting study. Upon removal of the mice expressing a CPA (see Materials and Methods), a comparison of the data obtained during the initial post-conditioning test from the remaining mice slated to receive intra-PFC VEH versus U0126 indicated no group difference in either CPP Score (VEH: 346.23 ± 70.29 s; U0126: 278.38 ± 48.71 s; t(15) = 0.81, *p* = 0.43) or distance traveled (VEH: 34.52 ± 2.56 m; U0126: 32.90 ± 3.20 m; t(15) = 0.34, *p* = 0.74) prior to the onset of microinjection procedures, with both VEH and U0126 mice exhibiting a robust MA-induced place-preference (data not shown; side effect: F(1,55) = 55.52, *p* < 0.0001; treatment effect and interaction, *p*’s > 0.25). 

Intra-mPFC administration of U0126 (0–100 nM) prior to the next four post-tests significantly attenuated CPP scores (Figure 2A) (Treatment effect: F(1,15) = 6.15, *p* = 0.03; Side effect: *p* = 0.56; interaction: F(3,45) = 2.59, *p* = 0.065). When infused with VEH, both groups exhibited a robust conditioned place-preference (one-sample t-tests, VEH-test1: t(5) = 3.00, *p* = 0.03; U0126-vehicle: t(10) = 4.65, *p* = 0.001) and their CPP Scores were superimposable (Figure 2A). Consistent with prior evidence that the expression of a MA-CPP persists in B6J males [26], VEH-pretreated mice continued to exhibit a robust MA-CPP across the remaining test sessions (one-sample *t*-tests, all *p*’s < 0.002). In contrast, a MA-CPP was absent in mice infused with U0126 (one-sample t-tests, all *p*’s > 0.32). Thus, inhibiting ERK signaling in the vmPFC blocks the expression of a MA-CPP. 

Despite its robust effects upon MA-CPP, U0126 infusion did not alter the locomotor activity expressed by the mice during testing (Figure 2B; Treatment X Dose ANOVA, all *p*’s > 0.20). Thus, the capacity of an intra-PFC infusion of U0126 to reduce the positive affective valence of MA was unrelated to non-selective effects upon locomotor activity. 

### 2.3. Transgenic Disruption of ERK-Dependent Phosphorylation of (T1123/S1126)mGlu5 Augments MA-Conditioned Reward

*Grm5^AA/AA^* mice exhibit increased alcohol-induced CPP but grossly perturbed cocaine-induced CPP [47]. Given the results above linking ERK activity to MA-conditioned reward, we examined the effects of disrupting ERK-dependent mGlu5 phosphorylation upon the capacity of repeated MA to elicit place-conditioning in a dose-dependent fashion. Interestingly, transgenic disruption of (T1123/S1126)mGlu5 phosphorylation produced a robust increase in the efficacy of MA to elicit a conditioned place-preference (Figure 3) (Side effect: F(1,68) = 30.00, *p* < 0.0001; Side X Genotype: F(1,68) = 18.12, *p* < 0.0001; no other main effects or interactions, *p*’s > 0.08), with the entire dose–response function of *Grm5^AA/AA^* mice shifted upwards of *Grm5^TS/TS^* controls. Thus, ERK-dependent phosphorylation of mGlu5 functions to suppress MA-conditioned reward in a manner similar to alcohol-conditioned reward [47].

In line with prior work [46,47], no genotypic differences were noted for the spontaneous locomotor activity expressed during the 15-min pre-conditioning or post-conditioning tests (*t*-tests, *p* = 0.18 and *p* = 0.58, respectively; data not shown). Thus, the effect of the *Grm5^AA/AA^* mutation upon MA-conditioned reward does not reflect alterations in spontaneous locomotor activity during testing. 

### 2.4. Transgenic Disruption of ERK-Dependent Phosphorylation of (T1123/S1126)mGlu5 Does Not Affect MA-Induced Locomotion or Locomotor Sensitization

*Grm5^AA/AA^* mice exhibit blunted cocaine-induced locomotor sensitization [46]. Thus, we examined for genotypic differences in MA-induced locomotor activity and for the change in locomotor activity across the four MA-conditioning sessions. Although it appeared that *Grm5^AA/AA^* were more sensitive to the acute locomotor-stimulating effect of 4 mg/kg MA (Figure 3B), an analysis of the dose-locomotor response function for acute MA (i.e., on the first conditioning session) did not detect any genotypic difference (Dose effect: F(3,74) = 4.00, *p* = 0.01; Genotype effect and interaction, *p*’s > 0.10). At all doses, MA-induced locomotor activity sensitized over the course of the four conditioning sessions (Figure 3C) (Dose effect: F(3,68) = 8.79; Injection effect: F(3,204) = 11.92, *p* < 0.0001; Dose X Injection: *p* = 0.26). However, the magnitude of the sensitization did not vary significantly as a function of genotype (Figure 3C) (no Genotype effect or interactions with the Genotype factor, *p*’s > 0.13), owing presumably to the high variability in extent to which MA sensitized responding. Thus, in contrast to our prior results for cocaine [46], ERK-dependent phosphorylation of (T1123/S1126)mGlu5 is not critical for either the psychomotor-activating or -sensitizing effects of MA. 

### 2.5. Transgenic Disruption of ERK-Dependent Phosphorylation of T1123/S1126-mGlu5 Robustly Attenuates Oral MA Reinforcement and Intake

*Grm5^AA/AA^* mice exhibit high levels of alcohol reinforcement and intake under operant-conditioning procedures [47]. Thus, we next determined in a new cohort of male and female *Grm5^TS/TS^* and *Grm5^AA/AA^* mice whether or not transgenic disruption of p(T1123/S1126)mGlu5 alters motivation for MA and MA intake when drug availability is contingent upon an operant response. 

### 2.6. Acquisition of Self-Administration

As reported previously by our group [28,48], when males and females were trained concurrently, no sex differences were observed for any measure during the five-day acquisition phase of the operant conditioning study (for active hole-pokes, for inactive hole-pokes, and for MA intake, Sex X Genotype X Day ANOVAs, no sex effects or interactions, *p*’s > 0.24). Thus, the data were collapsed across sex within each genotype. 

When presented with daily opportunities to nose-poke for reinforcement by 20 μL of a 10 mg/L MA solution, *Grm5^AA/AA^* mice exhibited lower active hole responding, relative to *Grm5^TS/TS^* mice, irrespective of the day of training (Figure 4A) (Day effect: F(4,140) = 14.13, *p* < 0.0001; Genotype: F(1,35) = 11.69, *p* = 0.002; Genotype X Day: *p* > 0.10). Likewise, *Grm5^AA/AA^* mice also exhibited lower inactive hole responding (data not shown; Genotype effect: F(1,35) = 4.19, *p* = 0.05; no Day effect or interaction, *p*’s > 0.25), suggesting lower overall behavioral output in the mutant mice. While both genotypes exhibited a progressive decline in their MA intake during training (Figure 4B) (Day effect: F(4,140) = 6.06, *p* < 0.0001)), mutant mice exhibited lower MA intake overall during initial training (Genotype effect: F(1,35) = 18.78, *p* < 0.0001; interaction: *p* = 0.29). Thus, interfering with ERK-dependent phosphorylation of mGlu5 reduces the initial reinforcing properties, and intake, of a low-dose 10 mg/L MA solution.

### 2.7. MA Demand under Increasing Response Requirement

When the response requirement for reinforcement by 10 mg/L MA was progressively increased across days, *Grm5^AA/AA^* mice maintained their lower level of active hole responding (Figure 4C) (Genotype effect: F(1,35) = 10.39, *p* = 0.003; interaction, *p* > 0.35) and similar genotypic differences were apparent also for inactive hole responding (data not shown; Genotype X Schedule: F(2,70) = 9.36, *p* < 0.0001). Consistent with prior observations in mice [28,48,49,50], MA intake dropped precipitously in *Grm5^TS/TS^* mice as a function of response requirement (Figure 4D) (Schedule effect: F(2,70) = 24.82, *p* < 0.0001); however, the MA intake by *Grm5^AA/AA^* mice was consistently less than that exhibited by *Grm5^TS/TS^* controls, although the genotypic difference was much more pronounced under the FR1 and FR2 scheduled of reinforcement (Figure 4D) (Genotype effect: F(1,35) = 23.89, *p* < 0.0001; Genotype X Schedule: F(1,70) = 9.96, *p* < 0.0001). Thus, mutation of (T1123/S1126)mGlu5 blunts the demand for MA, at least when 10 mg/L MA serves as the reinforcer. 

### 2.8. Dose-Response Function for Oral MA Reinforcement

Given the precipitous drop in MA intake under the FR5 schedule of reinforcement (Figure 4D), mice were returned to the original FR1 reinforcement schedule to examine for genotypic differences in the dose–response functions for MA reinforcement and intake. Examination of the dose–response function (5–40 mg/L MA) for active hole pokes indicated a shift downwards in *Grm5^AA/AA^* vs. WT mice (Figure 4E) (Genotype effect: F(1,35) = 11.02, *p* = 0.002; interaction, *p* > 0.31), and this genotypic difference was observed also in the cohort tested at the 80 mg/L dose (t(22) = 2.54, *p* = 0.02). Not shown, we observed no significant genotypic differences in the dose-dependent decline in inactive hole pokes observed during dose–response testing (5–40 mg/L MA: Dose effect: F(2,70) = 3.98, *p* = 0.02; Genotype effect and interactions, *p*’s > 0.10; 80 mg/L: t(22) = 1.98, *p* = 0.06), arguing that the genotypic differences in active hole pokes were selective for the MA-reinforced hole. 

Consistent with their results obtained during early and later acquisition of self-administration (Figure 4B,D), *Grm5^AA/AA^* mice continued to exhibit very low levels of oral MA intake during dose–response testing (Figure 4F). Although the magnitude of the genotypic difference in intake increased as a function of MA concentration (Figure 4F) (Genotype X Dose: F(2,68) = 14.42, *p* < 0.0001), significant group differences were detected at every MA concentration (*t*-tests, all *p*’s < 0.0001), with the largest genotypic difference in intake apparent in the cohort tested at 80 mg/L MA (Figure 4F, right) (t(22) = 20.53, *p* = 0.001). While these data argue that p(T1123/S1126)mGlu5 is necessary for MA reinforcement and intake under operant-conditioning procedures, based on the results from the place-conditioning study (Figure 3A), we propose that the blunted MA reinforcement and intake exhibited by *Grm5^AA/AA^* mice might reflect compensation for their heightened sensitivity to the primary rewarding properties of this drug.

### 2.9. Transgenic Disruption of ERK-Dependent Phosphorylation of (T1123/S1126)mGlu5 Blunts Oral MA Intake under DID Procedures

Another possibility to account for the discrepancy in findings between our place- and operant-conditioning experiments might relate to a genotypic difference in MA taste sensitivity. Thus, we next compared *Grm5^AA/AA^* and *Grm5^TS/TS^* mice for oral MA intake under DID procedures in a distinct cohort of MA-naïve female mice. To our surprise, no genotypic difference was detected for the dose–response function for MA intake in the home cage (Figure 5A) (Concentration effect: F(3,72) = 11.46, *p* < 0.0001; Genotype effect and interaction, *p*’s > 0.30) or for the average total MA consumed over the course of the study (Figure 5B; *t*-test, *p* = 0.30). Thus, the low MA intake exhibited by *Grm5^AA/AA^* mice does not appear to reflect a shift in MA taste sensitivity, or other factors associated with the capacity to drink MA. 

## 3. Discussion

The present study demonstrates that ERK hyperactivity within mPFC is a biochemical correlate of individual differences in the perception of MA’s interoceptive effects as rewarding. A directed neuropharmacological study provides causal evidence that mPFC ERK function is required for the positive affective valence of MA expressed by mice under place-conditioning procedures. Peculiarly, transgenic global disruption of ERK-dependent phosphorylation of (T1123/S1126)-mGlu5 increased sensitivity to MA’s positive affective properties, while markedly blunting MA reinforcement and intake under operant-conditioning procedures. The low MA intake exhibited by *Grm5^AA/AA^* mutants under operant-conditioning procedures cannot readily be accounted for by differential MA taste sensitivity, as no genotypic difference in home-cage MA-drinking was observed. Our findings are consistent with a key role for ERK hyperactivity within mPFC in driving the expression of MA-conditioned reward, but argue that (T1123/S1126)-mGlu5 within the mPFC is not likely the target involved. In contrast, ERK-dependent phosphorylation of (T1123/S1126)-mGlu5 is necessary for MA reinforcement, which may reflect actions within the mPFC or perhaps other mesocorticolimbic sites known to regulate MA self-administration behavior. 

### 3.1. Individual Differences in MA-Preference Are Positively Correlated with ERK Hyper-Activity within PFC

Similar to other addictive substances, a relatively small proportion of individuals who use MA recreationally develop MA Use Disorder. Over the past several years, we have employed an integrative approach to understand the neurobiological bases for individual differences in vulnerability to MA Use Disorder that includes the study of isogenic B6J mice spontaneously exhibiting MA-preference, -neutrality, or -aversion under place-conditioning procedures [26,27,50,51]. While we [27] and others, e.g., [42,52], have failed to detect an effect of repeated MA injections upon basal ERK phosphorylation within PFC, increased p-ERK expression is reported consistently in rodents expressing a MA-CPP (Figure 1; [52,53]), indicating that increased ERK activity within the mPFC is a response to the MA-associated environment, rather than MA administration per se—a finding consistent with an earlier MA-induced place-conditioning study in mice [51]. 

Providing our first piece of evidence that phospho-ERK expression within mPFC may drive MA-conditioned reward, herein, we identified p(Tyr204)ERK1/2 within the mPFC as a biochemical correlate of the affective valence of MA in male B6J mice (Figure 1D). Despite all mice receiving equivalent subchronic MA treatment as well as CPA and CPP mice exhibiting place conditioning effects of similar magnitude, an increase in both the total and relative expression of p(Tyr204)-ERK was observed only in mice expressing a CPP (Figure 1B,C), with CPA and neutral mice exhibiting similar levels of protein expression as MA-naïve, saline-conditioned controls (Figure 1C). These immunoblotting results for B6J males spontaneously expressing a MA-CPP align with the results of our earlier report in which we detected a modest increase in basal p(Tyr204)-ERK expression within the mPFC of male and female MA-naïve mice selectively bred for high MA-drinking (MAHDR), relative to their low-drinking (MALDR) counterparts [27]. Noteworthily, MAHDR exhibit, respectively, increased and decreased sensitivity to the conditioned rewarding and aversive properties of MA under place-conditioning procedures [22,24]. Taken all together, this body of correlational evidence implicates ERK hyper-activation within mPFC in driving the positive affective valence of MA and suggests drug cue/context-elicited ERK activation as at least one molecular contributor to the metabolic hyperactivity reported within the frontal cortex of individuals with MA Use Disorders in response to MA-associated stimuli, e.g., [54,55,56]. 

### 3.2. Local Inhibition of mPFC ERK Phosphorylation Eliminates an Established MA-CPP Phenotype 

Confirming the functional relevance of mPFC ERK for the expression of MA-conditioned reward, site-directed infusion of the MEK inhibitor U0126 reduced the magnitude of MA-CPP in B6J mice, with the lowest 1.0 nM dose effectively blocking CPP expression (Figure 2A). Notably, all of the mice in this study expressed a robust MA-CPP prior to microinfusion procedures, and thus we conclude that U0126 eliminated an established MA-CPP. Importantly, intra-mPFC U0126 infusion did not alter the locomotor activity of the mice at any point during testing (Figure 2B), indicating no off-target motor effects that might confound data interpretation. As reported previously by our group [26], the expression of MA-CPP is persistent in mice, as indicated by the stable conditioned approach response exhibited by control mice receiving daily VEH microinjections (Figure 2A). This finding for VEH controls, coupled with the fact that U0126 dosing was counter-balanced across test days, argues against potential factors associated with repeated microinjection (e.g., tissue damage) as a major contributing factor to our observed U0126 effects. 

ERK activation by MEK requires phosphorylation at both Thr202 and Tyr204, e.g., [57]. While we only assayed the Tyr202 phosphorylation site in our immunoblotting study (Figure 1C), the present U0126 data provide evidence that MEK-mediated activation of ERK within the mPFC is required for contextual control over MA-directed approach behavior. It remains to be determined whether or not intra-PFC infusion of U0126 during methamphetamine-conditioning prevents the development of the conditioned reward or aversion. However, our data extend to the mPFC, the results of earlier drug-induced place-conditioning studies in which systemic pretreatment with ERK inhibitors [58,59,60,61,62] or targeted ERK inhibition within the nucleus accumbens [51,63] were reported to inhibit drug-induced place-preference in rodents. The present data also align with other studies linking the efficacy of potential anti-addiction medications at blocking CPP with a reduction in phospho-ERK expression within mPFC, e.g., [53,64].

### 3.3. Grm5^AA/AA^ Mice Exhibit More Robust MA-CPP

Contrary to our MA-CPP results for U0126 (Figure 2A), *Grm5^AA/AA^* mice with disrupted ERK-dependent mGlu5 phosphorylation exhibit increased MA-CPP across an approximately 10-fold range of MA doses (Figure 3A). These results were not entirely unanticipated, as *Grm5^AA/AA^* mice also exhibit greater alcohol-induced CPP than *Grm5^TS/TS^* (WT) controls [47]. It is important to note, however, that unlike the present findings for MA-CPP (in which the entire dose–response function is shifted upwards in *Grm5^AA/AA^* mice versus WT controls; Figure 3A), the augmented alcohol-CPP exhibited by *Grm5^AA/AA^* mice reflects a shift to the right in the dose–response function for alcohol-CPP; *Grm5^AA/AA^* mice exhibit no conditioned response to low alcohol doses that effectively elicit a CPP in WT controls and an insensitivity to the conditioned aversive properties of higher alcohol doses [47]. Despite conditioning with doses as high as 4 mg/kg MA, we did not detect a high-dose MA-CPA in WT controls. However, it is interesting to note that *Grm5^AA/AA^* mice exhibit a robust MA-CPP at the lowest MA dose tested herein (0.5 mg/kg), while WT controls trended towards a CPA (Figure 3A). The blunted sensitivity to alcohol-CPA exhibited by *Grm5^AA/AA^* mice is attributed to ERK-dependent phosphorylation of mGlu5 within the bed nucleus of the stria terminalis (BNST) as the effects of an intra-BNST infusion of U0126 recapitulate the alcohol-CPP phenotype of *Grm5^AA/AA^* mice [47]. To the best of our knowledge, the role for the BNST in regulating the affective valence of MA is unexplored. Clearly, the opposite MA-CPP effects of intra-PFC U0126 infusion versus the T1123A/S1126A mutation of mGlu5 argue that the mPFC is not a neural locus involved in driving the MA-CPP phenotype of *Grm5^AA/AA^* mice. Based on our prior studies of alcohol-CPP in *Grm5^AA/AA^* mice [47], the BNST may play a key role in MA-CPP/CPA. Alternatively, accumulating evidence indicates a cross-sensitization between the rewarding properties of alcohol and MA [49,65,66] that we theorize reflects common glutamate-related neuroadaptations within the nucleus accumbens (see Ref. [26] for discussion). ERK-dependent signaling within the nucleus accumbens is required for the expression of amphetamine- and MA-CPP [51,63], raising the possibility that this brain region may be a potentially important site contributing to the MA-CPP phenotype of *Grm5^AA/AA^* mice. 

Although the neural loci contributing to the MA-CPP phenotype of *Grm5^AA/AA^* remain to be characterized, it should be noted that the heightened MA-CPP exhibited by *Grm5^AA/AA^* mice does not generalize to cocaine. In fact, both *Grm5^AA/AA^* and heterozygous *Grm5^TS/AA^* mice exhibit a very strong cocaine-CPA at doses that elicit a robust CPP in WT controls [47]. Further, the T1123A/S1126A mutation of mGlu5 abolishes cocaine-induced behavioral sensitization and neurochemical sensitization within the nucleus accumbens [46], the latter of which might contribute to their cocaine-averse phenotype, as cocaine-CPA is associated with blunted dopamine and glutamate sensitization within NAC [67]. Although the present study did not characterize the neurochemical phenotype of MA-treated *Grm5^AA/AA^* mice, we have reported that MA experience induces changes in extracellular glutamate and glutamate-related protein expression within the PFC and NAC that are not only distinct from, but sometimes opposite those produced by repeated cocaine treatment [26,29]. Whether or not the (T1123A/S1126A)-mGlu5 mutation also affects MA-induced glutamate plasticity within the NAC, mPFC, or other addiction-relevant brain regions (e.g., BNST) is an important next step to account for the polar opposite effects of the (T1123A/S1126A)-mGlu5 mutation upon the affective valence of cocaine versus MA and alcohol. 

### 3.4. Grm5^AA/AA^ Mice Exhibit Reduced Oral MA Self-Administration

From the outset of operant-conditioning procedures, *Grm5^AA/AA^* mice exhibited lower MA reinforcement and intake, compared to *Grm5^TS/TS^* controls (Figure 4). Although *Grm5^AA/AA^* also exhibit lower sucrose self-administration under operant-conditioning procedures, they are reported to exhibit higher alcohol reinforcement and intake versus WT controls [47]. Thus, the low-level MA self-administration observed herein does not likely reflect an operant-learning, motivational, or motor deficit secondary to the mutation. Moreover, as the amount of MA consumed during training was relatively low (<0.5 mg/kg MA/day), it is also not likely that the low level of responding/intake reflects the induction of stereotyped behavior that might interfere with the operant response. While we did not monitor motor activity during self-administration procedures, the fact that no genotypic difference in MA-induced locomotion was detected during place-conditioning procedures (Figure 3B,C) when mice were repeatedly administered MA doses well in excess of the doses consumed during operant-conditioning also argues against differential sensitivity to MA’s psychomotor effects as contributing to the markedly blunted MA reinforcement exhibited by *Grm5^AA/AA^* mice. 

The genotypic difference in MA-reinforced responding was apparent from the second day of training under an FR1 schedule of reinforcement and persisted throughout training (Figure 4A). As reported previously in female and male B6J mice [48,49,50], MA-directed responding does not vary much with increasing response requirement (Figure 4C), but the intake of low-concentration MA drops precipitously with increasing demand (Figure 4D). Such findings argue that mice do not find low-concentration MA particularly reinforcing. As B6J mice consume less MA than mice on other genetic backgrounds (e.g., DBA2/J) [68,69], it remains to be determined whether or not the low MA demand and intake observed in our studies to date reflect the B6J genetic background of the mice or the relatively low training doses employed in our studies. Nevertheless, our procedures are sufficient to detect robust genotypic differences in both MA-directed responding and intake [28,48] (Figure 4C,D), to further the notion that ERK-dependent phosphorylation of mGlu5 is necessary for MA reinforcement. 

*Grm5^AA/AA^* mice exhibit consistently lower MA-directed responding and intake than WT controls across a greater than 10-fold range of MA concentrations (5–80 mg/L). In fact, the genotypic difference in daily MA intake widens as a function of the MA reinforcer concentration (Figure 4F), and this genotype X MA dose interaction is apparent also in our study of MA drinking in the home cage (Figure 5A). The consistently low levels of MA-directed responding exhibited by *Grm5^AA/AA^* mice are indicative of lower MA reinforcement, which is difficult to reconcile with their more robust MA-CPP phenotype (Figure 3A). Differential bitter tastant sensitivity is unlikely to be a factor contributing to the lower intake of MA during behaviorally contingent access, because no genotypic differences were apparent under home-cage drinking procedures (Figure 5A). Given the robust MA-CPP phenotype of the *Grm5^AA/AA^* mice, it is possible that their low level of MA reinforcement may reflect a compensation for their increased sensitivity to MA’s rewarding properties. 

## 4. Materials and Methods

### 4.1. Subjects 

Adult (eight weeks of age) male inbred C57BL/6J (B6J) mice were purchased from Jackson Laboratories (Sacramento, CA, USA), and housed in groups of four in a temperature (25 °C)- and humidity (71%)-controlled colony room, under a 12-h reverse light cycle (lights off: 10:00 h). Heterozygous breeder pairs of transgenic mice with alanine substitution mutations at T1123 and S1126 sites on mGlu5 were obtained from the laboratory of Dr. P.F. Worley (Johns Hopkins University School of Medicine, Baltimore, MD, USA) (generated as described in Refs. [46,70]) and used to generate *Grm5^TS/TS^* and *Grm5^AA/AA^* offspring. Mice bred in-house remained multi-housed with same-sex littermates throughout experimentation. All studies of mutant mice employed both male and female littermates from at least three different litters, with each experimental condition consisting of approximately equal numbers of male and female mice. As no sex difference or interactions with sex were detected during statistical analyses, the data for males and females are collapsed for presentation. The *Grm5^TS/TS^* and *Grm5^AA/AA^* mice were tested between seven and 15 weeks of age, and food and water were available ad libitum throughout the entire study unless otherwise indicated. The experiments followed a protocol consistent with NIH guidelines presented in the recently revised Guide for Care and Use of Laboratory Animals (2014) and approved by the IACUC of the University of California, Santa Barbara, CA, USA (protocol numbers 829.1, approved 8 May 2014 and 829.2, approved 27 March 2017).

### 4.2. Place-Conditioning and Locomotor Activity 

The MA place-conditioning procedures employed in the present experiments were identical to those described previously by our group [26,27,28,48,49,50] and are outlined in the inset of Figure 1. Place-conditioning commenced with a pre-conditioning test in which animals were allowed to freely explore both compartments of the place-conditioning chambers in a drug-free state to habituate them to the apparatus. Overall, there was no preference for one compartment versus the other during this pre-test, indicating that the apparatus, itself, was unbiased. Mice were then injected intraperitoneally (IP) with 2 mg/kg MA (vol = 10 mL/kg) and randomly confined to one compartment and, on alternating days, were injected with saline and confined to the alternate compartment for a total of 4 pairings each. The day following the last conditioning session, a post-conditioning test was conducted in which animals were again allowed free access to both chambers in a drug-free state, and the difference in the time spent in the MA- versus saline-paired compartment (i.e., CPP Score in sec) was used to index the direction and magnitude of the place-conditioning. 

As detailed previously [26,27,71], for our examination of the ERK correlates of MA place-conditioning in C57BL/6J mice, we removed brain tissue within 2–5 min of test completion. Mice exhibiting a CPP Score > +100 s were phenotyped as “MA-preferring” (CPP), mice exhibiting a CPP Score < −100 s were phenotyped as “MA-avoiding” (CPA), and mice exhibiting CPP Scores between −100 and +100 s were phenotyped as “MA-neutral” (Neutral). 

For the study of ERK-dependent regulation of MA-conditioned reward, only CPP and neutral mice were included, as conducted in prior neuropharmacological studies by our group [26,27,71]. In all, six out of the original 32 mice involved in the neuropharmacological study that exhibited a CPA (i.e., 16%) did not undergo microinjection procedures. This proportion of CPA mice is consistent with recently published work [26,50], indicating that a relatively small subpopulation of B6J mice perceive the interoceptive effects of 2 mg/kg as aversive. The remaining B6J mice were then slated to receive intra-PFC infusions of either vehicle (VEH) or various doses of the ERK inhibitor U0126, immediately prior to each subsequent post-conditioning test (see below for details). 

The MA-induced place-conditioning phenotype of *Grm5^AA/AA^* mice was determined using a between-subjects design in which different groups of mice of both sexes were conditioned with 0.5, 1, 2, or 4 mg/kg MA to establish a dose–response function. In our experience [28,48,49,50] and as reported by others [23,24], this dose-range is sufficient to detect group differences in the affective valence, as well as locomotor-stimulating effects, of MA. With the exception of dosing, the place-conditioning procedures employed in our study of *Grm5^TS/TS^* and *Grm5^AA/AA^* mice were identical to those employed in our study of B6J mice.

Each place-conditioning session was 15 min in duration, and throughout conditioning and testing, the total distance traveled was also recorded to examine for off-target effects of U0126 infusion and genotypic differences in MA-induced psychomotor activation and sensitization [26,27,28,48,50,65,66,71,72].

### 4.3. Operant-Conditioning for Oral Methamphetamine Reinforcement

To examine the effects of the *Grm5^AA/AA^* mutation upon MA reinforcement and intake, we trained female and male *Grm5^AA/AA^* (*n* = 23; 12 females and 11 males) and *Grm5^TS/TS^* (*n* = 14; seven females and seven males) mice to nose-poke for oral MA using operant-conditioning procedures similar to those described in recent studies by our group [26,28,48,49,50,52]. An outline of the operant-conditioning procedures is provided in the inset of Figure 4. Mice were placed into standard mouse operant-conditioning chambers (MedAssociates, St Albans, VT, USA), equipped with two nose-poke holes with a liquid receptacle between them. One hole (the “active”, MA-associated hole) delivered 20 μL of the reinforcer from an infusion pump when activated, along with the simultaneous presentation of a 20-s tone/light compound stimulus. The MA reinforcer was dissolved in potable tap water at the concentrations denoted below. During the 20-s activation period, further nose-pokes in the active hole were recorded but did not result in any programmed consequences. Nose-pokes in the opposite hole (the “inactive”, MA-unpaired hole) had no programmed consequences but were recorded to determine reinforcer efficacy. Chambers were ventilated and sound-attenuated. Conditioning began with a five-day training period on an FR1 reinforcement schedule (one nose-poke/reinforcer) using a 10 mg/L MA reinforcer, after which mice were removed from the study if they failed to allocate 70% of their total responding towards the active hole and/or failed to emit at least 10 active nose-pokes during the 1-h session. The mice that met both acquisition criteria were then tested for reinforcement by 10 mg/L MA under an FR2 and then an FR5 schedule (two nose-pokes/reinforcer and five nose-pokes/reinforcer, respectively) for five days each. Reinforcement schedule was then dropped back to FR1 and reinforcer concentrations switched to 5 mg/L, 20 g/L, and 40 mg/L, each for five days, to determine genotypic differences in the dose–response function for MA reinforcement. Initially, our IACUC limited the maximum MA reinforcer concentration to 40 mg/L. However, having demonstrated that this concentration clearly lay on the ascending limb of the dose-intake function (see below; see also Refs. [26,49,50]), we were allowed to assay a distinct cohort of mice at 80 mg/L MA. For this additional cohort, *Grm5^AA/AA^* and *Grm5^TS/TS^* mice (*n* = 11/genotype) were trained to self-administer 10 mg/L MA under the FR1 training schedule and then tested only at the 80 mg/L dose on the same FR1 schedule. 

To determine individual MA intake for each session, mice were returned to their home cages, and the volume of reinforcer left in the liquid receptacle of each chamber was measured by pipetting. The volume remaining was subtracted from the total volume delivered (i.e., number of reinforcers × 20 µL) and then expressed as a function of body weight to determine MA intake (on a mg/kg body weight basis). Body weight was determined weekly. For the mice undergoing dose–response testing, the average number of active nose-pokes, the ratio of active vs. total nose-pokes, and MA intake were analyzed by a mixed ANOVA, with repeated measures on the factors of Training Day, Reinforcement Schedule, or MA Dose. The data from the cohort of mice that was tested at the 80 mg/L were analyzed using a *t*-test across the Genotype factor. Two-tailed Pearson correlational analyses were also conducted to relate dependent measures with CPP Score (α = 0.05). 

### 4.4. Methamphetamine Intake under Drinking-in-the-Dark Procedures

To examine the effects of the *Grm5^AA/AA^* mutation upon MA intake under behaviorally noncontingent drug-access procedures, we employed a modified version of the drinking-in-the-dark (DID) procedure, in which female *Grm5^TS/TS^* (*n* = 14) and female *Grm5^AA/AA^* (*n* = 12) mice were presented simultaneously with sipper tubes containing 5, 10, 20, and 40 mg/L MA solutions and allowed to drink for a 2-h period (see inset in Figure 5). Only females were employed in this study due to a limited number of male subjects at the time of testing. Females are reported to consume more oral MA than males under certain operant-conditioning procedures [25,48,50], although a sex difference in MA-drinking in the home cage has not been reported in the extant literature, e.g., [13,22]. As conducted in a prior studies of MA-drinking in B6J mice [26] and alcohol-drinking in *Grm5^TS/TS^* versus *Grm5^AA/AA^* mice [47], mice were single-housed in drinking cages lined with sawdust bedding for a minimum of 1 h to habituate to the drinking environment. MA bottles were presented for a 2-h period, beginning at 3 h into the dark phase of the circadian cycle. Upon completion of the 2-h drinking session, the mice were returned to their home cages, the difference in bottle weight before and after the drinking session was used to determine the volume consumed from each solution, and the data were expressed as the mg of MA consumed per kg body weight. Mice underwent MA-DID procedures for a period of seven days, and the average MA intake under DID procedures were analyzed using a Genotype X Concentration ANOVA, with repeated measures on the Concentration factor. 

### 4.5. Stereotaxic Surgery, U0126 Microinjection, and Histology 

B6J mice slated for microinjection underwent stereotaxic surgery to implant stainless steel guide cannulae (7 mm, 26 gauge; Eagle Stainless; Warminster, PA, USA) above the mPFC using procedures identical to those described in our prior work, e.g., [27,71,73]. All surgeries were performed under isoflurane anesthesia (1.5–2%), using oxygen as the carrier gas. Once anesthetized, a mouse was placed in a Kopf stereotaxic device, and its head was stabilized with tooth and ear bars. The skull was then exposed and leveled. Holes were drilled based on coordinates from Bregma (AP: +1.8 mm, ML: ±0.5 mm; DV −1.0 mm), according to the Paxinos and Franklin (2007) mouse brain atlas. The guide cannulae were then lowered to 2 mm above the mPFC and were fixed in place with light-cured dental resin. Surgical incisions were closed, using tissue adhesive as necessary. Dummy cannulae (30 gauge; length equivalent to guide cannulae) were inserted into the guide cannulae to reduce externalization. Animals were administered the non-steroidal, anti-inflammatory banamine (2 mg/kg, SC), once during the surgical procedure and then twice a day for the first 48 h post-operation. Animal health was monitored daily following surgery, and all mice were allowed at least five days recovery prior to commencing place-conditioning procedures as described above. Following MA-conditioning, a post-test was conducted, prior to which none of the mice received any intracranial treatment. This was done to (1) ensure equivalent conditioned behavior and locomotor activity across the pretreatment groups prior to microinjections and (2) to identify and remove from the study any mice expressing a CPA. 

As outlined in the inset of Figure 2, a within-subjects design was employed to assay the effects of an intra-mPFC infusion of varying doses of the MEK1/2 inhibitor U0126 (0–100 nM) [74] upon the expression of the MA-conditioned response. U0126 inhibits MEK1/2-dependent activation of ERK [74], and U0126 doses were selected based on data from our laboratory indicating their effectiveness at altering alcohol consumption when infused intracranially in B6J mice [47]. Approximately 5–7 min prior to the post-test, dummy cannulae were removed, and microinjectors (30-gauge, 9 mm long) were lowered bilaterally through the guide cannulae into the mPFC. A 1 μM stock solution of U0126 was prepared by dissolving in water and aliquots stored at −80 °C until use. Fresh U0126 working solutions were prepared daily by diluting the stock with sterilized water for injection and then infused at a rate of 0.5 μL/min for 1 min (vol/side = 0.5 μL). As conducted in prior neuropharmacological studies of MA-induced place-conditioning [26,27,71], microinjectors were left in place for an additional 1 min, slowly removed, and sterilized dummy cannulae re-inserted into the guides. Animals then were placed into the place-conditioning chambers for the 15-min test period. The order of U0126 dosing was pseudo-randomly assigned across four consecutive test days, with 1–3 mice per cohort receiving any particular U0126 dose. An additional subset of mice within each of the three experimental cohorts tested under our microinjection procedures (*n* = 1–3/cohort) were infused only with water vehicle (VEH) to control for the effects of repeated microinjection upon place-preference magnitude. The data from the U0126 study were analyzed using a mixed ANOVA with repeated measures on the Dose factor.

Upon the completion of testing, mice were euthanized and brains drop-fixed in 4% paraformaldehyde. Brains were sectioned (50 μm) along the coronal plane on a vibratome and then stained with Cresyl violet. Sections were then viewed under a light microscope for microinjector placement within the vmPFC. Mice with placements outside the prelimbic/infralimbic cortices were excluded from the data analyses. 

### 4.6. Immunoblotting 

Immunoblotting procedures were conducted to determine whether or not individual differences in the magnitude and direction of MA-induced place-conditioning exhibited by B6J mice correlated with ERK activity within the mPFC by examining for total and (Tyr204)ERK phosphorylation (pERK). As described in our published work using these same mice [26,27,71], a subset of CPP, CPA, and Neutral mice from the place-conditioning study (*n* = 12–14/phenotype) were rapidly decapitated immediately (within 2–5 min) upon completion of the post-test and phenotyping. The mPFC was excised over ice and protein content in the samples determined using the BCA method, as described previously (see Figure 1B) [27]. To provide a baseline of ERK expression and phosphorylation, a MA-naïve control group was included in this study that received four saline injection pairings with each compartment of the place-conditioning apparatus and then allowed free-access to both compartments [26,27,71]. Immunoblotting procedures for the detection of ERK in 30 µg protein samples were performed as described previously [27,47,57,65] and employed a rabbit anti-ERK1/2 primary antibody (Santa Cruz Biotechnology, Santa Cruz, CA, USA; 1:1000 dilution) or a mouse p(Tyr204)ERK1/2 (Santa Cruz Biotechnology, Santa Cruz, CA, USA; 1:1000 dilution). After primary antibody incubation, membranes were washed prior to being incubated with a horseradish peroxidase-conjugated goat anti-rabbit or anti-mouse secondary antibody (Millipore, Burlington, MA, USA; 1:40,000–1:80,000 dilution) for 90 min. Membranes were then washed again, and immunoreactive bands were detected by enhanced chemiluminescence using either ECL Plus (GE Healthcare, Chicago, IL, USA) or Pierce SuperSignal West Femto (Fisher Scientific, Waltham, MA, USA). A rabbit anti-calnexin polyclonal primary antibody (Stressgen) was used to ensure even protein loading (30 µg/lane) and transfer. Scans of the raw immunoblots employed in this analysis are provided as unpublished material. ImageJ (NIH, Bestheda, MD, USA) was used to quantify the immunoreactivity of each protein. pERK/total ERK ratios were calculated for each animal, and for group comparisons, values of MA-conditioned animals were expressed as a percentage of average of the saline-injected animals on each gel (*n* = 3–4). The data were analyzed by ANOVA, followed by LSD post-hoc tests and a Pearson test correlated the ratio of phospho:total ERK with CPP Score (α = 0.05 for analyses).

## 5. Conclusions

The results of the present study provide novel evidence that ERK activation within mPFC is a biomolecular response to a drug-conditioned context that drives the positive affective effects of MA. While the downstream targets of ERK phosphorylation within the mPFC driving MA-conditioned reward are unknown, mGlu5 appears to be a critical target of ERK-directed signaling regulating MA reinforcement and intake.

## Figures and Tables

**Figure 1 ijms-22-01473-f001:**
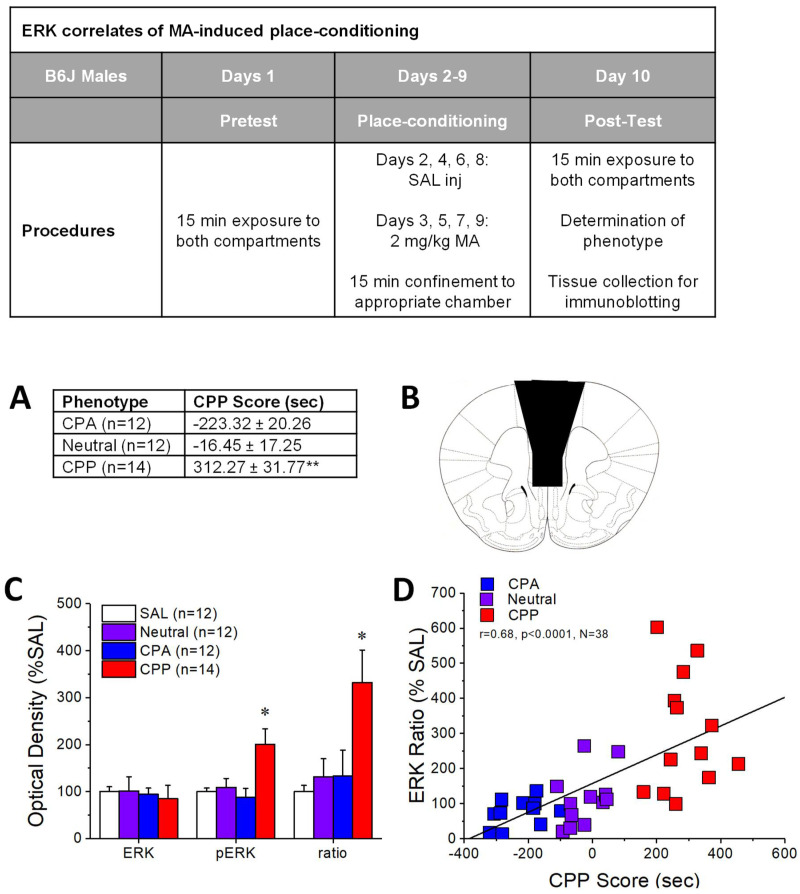
Elevated phospho-ERK expression within mPFC is a biochemical correlate of the affective valence of methamphetamine. (Inset): Outline of the procedural timeline for the immunoblotting study of ERK correlates of the place-conditioning elicited by the repeated pairing of 2 mg/kg methamphetamine (MA) in male C57BL/6J mice. Mice were tested for the expression of their conditioned response in a MA-free state (15 min session) and brains removed (~2–5 min later) for immunoblotting. (**A**) Summary of the differences in the time spent in the methamphetamine (MA)—versus saline (SAL)—paired compartment of the place-conditioning apparatus between mice spontaneously exhibiting a MA-conditioned place-preference (CPP), a MA-conditioned place-aversion (CPA), and no conditioned response (Neutral). Sample sizes are indicated in parentheses. Note data in Panel A is from Ref. [26]. (**B**) Cartoon depicting the tissue dissection of the mPFC. (**C**) Summary of the mPFC levels of total ERK, p(Tyr204)-ERK, and their ratio exhibited by CPP, Neutral, and CPA mice, as well as mice conditioned with SAL. The raw immunoblots are provided in the unpublished materials. (**D**) Results of the correlational analysis conducted between the CPP Score and the ratio of phosphorylated to total ERK within the mPFC. The data in Panels (**A**,**C**) represent the mean ± SEMs of the number of mice indicated in parentheses. * *p* < 0.05 vs. all other groups; ** *p* < 0.05 vs. CPA and Neutral.

**Figure 2 ijms-22-01473-f002:**
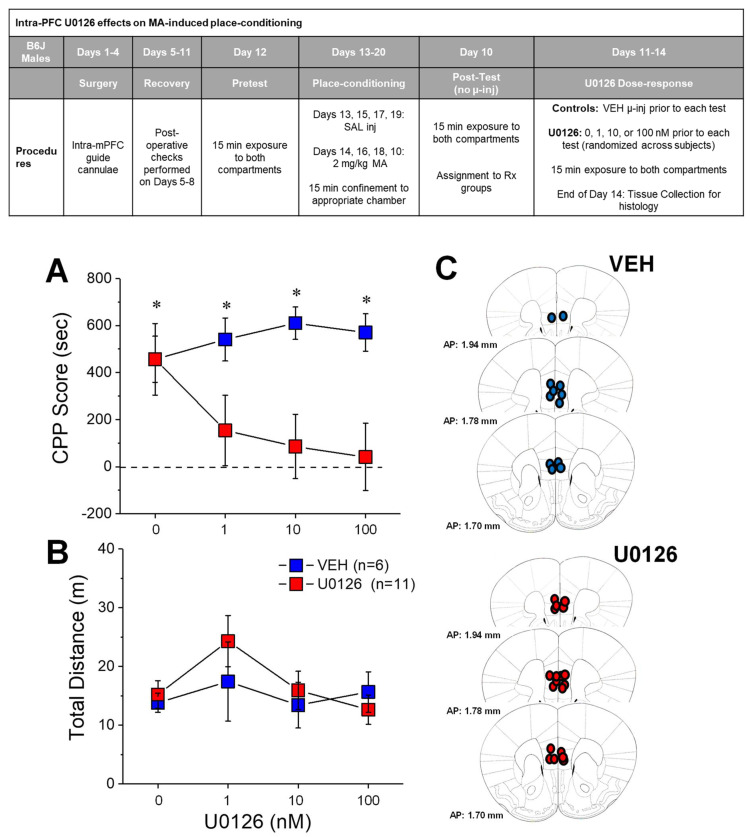
ERK inhibition within the mPFC blocks an established MA-conditioned place-preference, without affecting locomotor activity. (Inset) Procedural time-line for the neuropharmacological study of ERK inhibition upon the expression of a MA-conditioned place-preference. (**A**) Intra-mPFC infusion of the MEK inhibitor U0126 lowered the expression of a place-preference induced by the repeated pairing of 2 mg/kg methamphetamine. In contrast, no effect of repeated vehicle (VEH) infusion was detected. (**B**) U0126 infusion did not alter the locomotor activity of the mice during place-preference testing. (**C**) Cartoons depicting the locations of the microinjector tips within the mPFC. The data represent the means ± SEMs of the number of mice indicated in parentheses. * *p* < 0.05 CPP Score vs. 0 (i.e., presence of a conditioned response; 1-sample *t*-tests).

**Figure 3 ijms-22-01473-f003:**
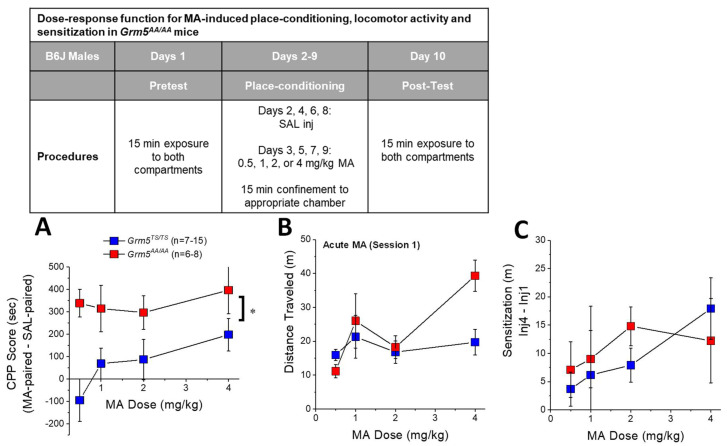
Disruption of ERK-dependent phosphorylation of mGlu5 augments methamphetamine-induced place-preference without affecting methamphetamine-induced psychomotor activity. (Inset) Procedural timeline of the study characterizing the effects of the *Grm5^AA/AA^* mutation upon MA-induced changes in behavior under place-conditioning procedures. Comparison of the dose–response functions for (**A**) methamphetamine (MA)-induced place-preference, (**B**) acute drug-induced locomotor activity, and (**C**) locomotor sensitization (defined as the difference in distance traveled from injection 1 to 4 of MA-conditioning) between WT *Grm5^TS/TS^* and mutant *Grm5^AA/AA^* mice. The data represent the means ± SEMs of the number of mice indicated in parentheses. * *p* < 0.05 (main effect of Genotype).

**Figure 4 ijms-22-01473-f004:**
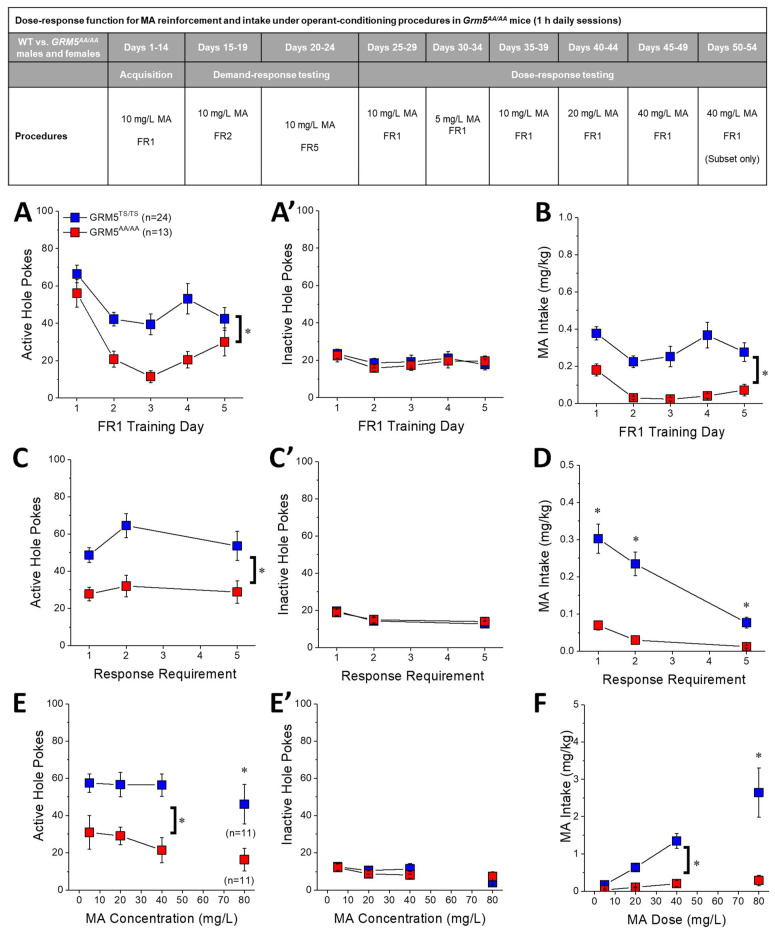
Disruption of ERK-dependent phosphorylation of mGlu5 blunts methamphetamine reinforcement and intake under operant-conditioning procedures. (Inset) Procedural timeline of the operant-conditioning studies conducted in *Grm5^AA/AA^* and *Grm5^TS/TS^* mice. When the first five days of self-administration training were considered, *Grm5^AA/AA^* mutants exhibited less (**A**) methamphetamine (MA)-directed responding for reinforcement by 20 mg/L MA and (**B**) less MA intake than WT *Grm5^TS/TS^* mice. When the response requirement for reinforcement by 20 mg/L MA progressively increased, mutant mice continued to exhibit (**C**) less MA-directed responding and (**D**) less intake. When the dose of the MA reinforcer varied under an FR1 schedule of reinforcement, the MA-directed responding (**E**) and MA intake (**F**) continued to be lower in *Grm5^AA/AA^* mice versus *Grm5^TS/TS^* controls. No genotypic differences were detected for responding in the inactive hole during (**A’**) early training, (**C’**) demand testing, or (**E’**) dose–response testing. The data represent the means ± SEMs of the number of mice indicated in parentheses. Note: not all mice tested for reinforcement by 5–40 mg/L MA advanced to testing under the 80 mg/L dose. * *p* < 0.05 (main Genotype effect); * *p* < 0.05 vs. *Grm5^TS/TS^* (*t*-test).

**Figure 5 ijms-22-01473-f005:**
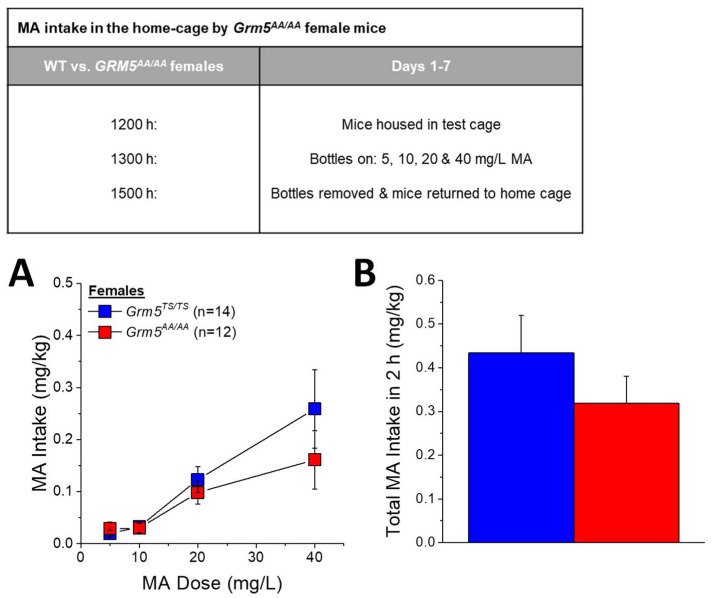
Disruption of ERK-dependent phosphorylation of mGlu5 only modestly blunts MA intake in the home cage. When presented simultaneously with four sipper tubes containing, 5, 10, 20, and 40 mg/L MA, (**A**) *Grm5^AA/AA^* mice tended to consume less 40 mg/L MA and (**B**) exhibited lower overall MA intake than their WT *Grm5^TS/TS^* controls. Neither of the genotypic differences were statistically reliable. The data represent the means ± SEMs of the number of mice indicated in parentheses.

## Data Availability

The data presented in this study are available on request from the corresponding author. The data are not publicly available due to formatting constraints.

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
