# Peer review of "ERK-Directed Phosphorylation of mGlu5 Gates Methamphetamine Reward and Reinforcement in Mouse"

_ijms, 2021, doi:10.3390/ijms22031473_

Round 1
Reviewer 1 Report
The manuscript entitled, “ERK-directed phosphorylation of mGlu5 gates methamphetamine reward and reinforcement in mouse” describes 1) how methamphetamine-induced conditioned place preference (MA-CPP) correlates with phosphorylated ERK in the mPFC, 2) how inhibition of ERK phosphorylation in the mPFC removes the expression of MA-CPP, and 3) how the transgenic mice with globally mutated mGlu5 receptors which cannot be phosphorylated by ERK show enhanced MA-CPP but reduced responding for MA in operant self-administration tests.
The results are of interest, experiments appear to be carefully performed and the conclusions drawn from the results sound. However, it was difficult to follow the design of experimentation such as what mouse cohorts and time points were used in different tests. A figure with a design of experiments would help.
In the discussion (lines 336-378), it could also be commented on what would happen if ERK phosphorylation were inhibited during MA-conditioning phases by injections of U0126 to the mPFC. Has anyone performed such CPP experiments with any addictive drugs? Would such experiments clarify why the mutations disrupting phosphorylation of mGlu5 receptor led to enhanced MA-CPP.
In the abstract, the methods used should be explained in more detail (conditioned-place preference/aversion, operant self-administration).
Fig.1A table: Are these the same mice as in Ref 26 (as indicated on line 90) or a subset of those. Please indicate n for each group in A. Are mice in C a subset of mice in A.
Fig.1, figure legend: Please correct, of Table 204. Mention the dose of MA used for CPP, and the timing for collecting the brains for phosphoERK analysis.
Line 99: Consider reformulating this sentence. Is it not known at this point whether the expression of MA-place preference induces ERK phosphorylation or ERK phosphorylation leads to the expression of MA-CPP.
Line 109: What is this distinct group of B6J mice? Are they included in Fig.1A table? Are they subset of mice in Fig.1A table? Are “neutral” mice included?
Figure 2. Please indicate in the figure legend the dose of MA used for conditioning phase.
Lines 137-139: Please correct or remove.
Line 154: GRMAA/AA, please correct.
Line 370: PPA adolescents?
Lines 422-424: Would use of much lower doses of MA, rather than higher, in self-administration study elucidate the question whether the mutated mGlu5 mice are more sensitive to its rewarding effects?
Line 428: correct Materials
Line 446-452: It is not clear whether unbiased or biased place conditioning was used. And if biased, whether MA assigned compartment was preferred or non-preferred. Please explain this in more detail.
Line 459-466: Please clarify. What do you mean,…as conducted in prior neuropharmacological studies by our group. What are these 32 mice, out of which 6 showed CPA, in Fig.1 table there are 38 mice out of which 12 exhibited CPA. Are these both groups (Fig.1 and Fig.2) from the same large cohort of mice used already in Ref. 26, and then divided to subgroups and subsequent tests. Was stereotaxic surgery performed before or after MA-place conditioning? A schematic figure would help to understand the study design.
Line 519-520: Was water available during this 2 hours?
References are indicated by numbers otherwise, but in the Discussion by names, are these references included in the Reference list?
Author Response
Reviewer 1
The results are of interest, experiments appear to be carefully performed and the conclusions drawn from the results sound. However, it was difficult to follow the design of experimentation such as what mouse cohorts and time points were used in different tests. A figure with a design of experiments would help.
Reply: We thank the reviewer for their kind opinion of our report. We have now included procedural time-lines as insets for each of the figures to facilitate visualization of study design.
In the discussion (lines 336-378), it could also be commented on what would happen if ERK phosphorylation were inhibited during MA-conditioning phases by injections of U0126 to the mPFC. Has anyone performed such CPP experiments with any addictive drugs? Would such experiments clarify why the mutations disrupting phosphorylation of mGlu5 receptor led to enhanced MA-CPP.
Reply: As summarized in the Discussion of the original version of the article, some studies have examined the effects of daily pretreatment with ERK inhibitors on the development of drug-induced place-preference. Due to revisions, these citations are now located on line 322. For instance, Valjent and colleagues (2000) showed that daily pretreatment with SL 327, prior to cocaine administration and conditioning, blocked the development of a cocaine-conditioned place-preference. We have not located any reports pertaining to the development of methamphetamine-induced conditioning and we have included a sentence to that effect in lines 328-330.
In the abstract, the methods used should be explained in more detail (conditioned-place preference/aversion, operant self-administration).
Reply: We have provided more experimental detail in the abstract as suggested.
Fig.1A table: Are these the same mice as in Ref 26 (as indicated on line 90) or a subset of those. Please indicate n for each group in A. Are mice in C a subset of mice in A.
Reply: Yes, these mice are the same as those employed in the immunoblotting studies of Ref 26 (and Ref. 27). It is for this very reason why we did not include a graphical depiction of their behavior as the data is graphed in these other reports. The sample sizes for these groups was provided in Fig.1C, but we have now included them also in the table in Fig.1A.
Fig.1, figure legend: Please correct, of Table 204. Mention the dose of MA used for CPP, and the timing for collecting the brains for phosphoERK analysis.
Reply: We have made corrections as suggested. We have also mentioned in the main text that tissue was dissected within 2-5 min post-testing.
Line 99: Consider reformulating this sentence. Is it not known at this point whether the expression of MA-place preference induces ERK phosphorylation or ERK phosphorylation leads to the expression of MA-CPP.
Reply: We agree with the reviewer, which is why we only referred to a correlational, not causal, relationship between p-ERK expression and behavior on Line 99.
Line 109: What is this distinct group of B6J mice? Are they included in Fig.1A table? Are they subset of mice in Fig.1A table? Are “neutral” mice included?
Reply: We apologize for the confusion. The term distinct was employed to indicate that the C57BL/6J mice employed in the microinjection experiment were distinct from those in the immunoblotting experiment (as the mice were killed at the end of the immunoblotting study). As such, they are not included in Fig.1A. We have now clarified this in lines 121-122 of the revised manuscript. As described in the original report in lines 459-466, we only included CPP and Neutral mice in the microinjection study as the goal was to determine the role for ERK in the expression of a place-preference (see lines 468-475 of revised report). As is apparent from our Results, when the data for CPP and Neutral mice are averaged, the mean CPP Score is above 250 sec on the MA-paired side and this strong place-preference is maintained in mice infused daily with vehicle.
Figure 2. Please indicate in the figure legend the dose of MA used for conditioning phase.
Reply: We have included the doses in the figure legend.
Lines 137-139: Please correct or remove.
Reply: The odd text in the figure legend for Fig.2 reflected editorial errors during the conversion of our report from its original Word document and have been corrected. Thank you for noticing them.
Line 154: GRMAA/AA, please correct.
Reply: Thank you. Corrected.
Line 370: PPA adolescents?
Reply: We humbly apologize but a premature version of the report was originally submitted and the citations within the Discussion had not been converted to numbers. We have thoroughly reviewed the Discussion and now all citations are included in the Reference section, numbered,
Lines 422-424: Would use of much lower doses of MA, rather than higher, in self-administration study elucidate the question whether the mutated mGlu5 mice are more sensitive to its rewarding effects?
Reply: This study included very, very low doses of MA and it is clear from the data that the most robust genotypic difference is observed at 80 mg/L. This warrants follow-up investigation using higher MA doses that lie closer to the peak of the intake function and, preferably, lie on the descending limb of the intake function so we can have a more thorough understanding of how the mutation is affecting MA sensitivity. Unfortunately, our IACUC was (and continues to be) very hesitant about allowing animal access to higher concentrations, despite their ability to control their own intake (and opt NOT to drink the MA solution) and we hope to be able to convince them that a full characterization of the dose-response function for MA-taking is critical to our ability to interpret our findings.
Line 428: correct Materials
Reply: That was a typo from the file conversion. Corrected.
Line 446-452: It is not clear whether unbiased or biased place conditioning was used. And if biased, whether MA assigned compartment was preferred or non-preferred. Please explain this in more detail.
Reply: We have now indicated that both our apparatus and our conditioned procedures were unbiased (see lines 455-457).
Line 459-466: Please clarify. What do you mean,…as conducted in prior neuropharmacological studies by our group. What are these 32 mice, out of which 6 showed CPA, in Fig.1 table there are 38 mice out of which 12 exhibited CPA. Are these both groups (Fig.1 and Fig.2) from the same large cohort of mice used already in Ref. 26, and then divided to subgroups and subsequent tests. Was stereotaxic surgery performed before or after MA-place conditioning? A schematic figure would help to understand the study design.
Reply: When we conducted the place-conditioning procedures, we examined behavior prior to the U0126 infusion. Of the 32 mice that underwent place-conditioning, 6 mice exhibited a CPA. We wanted a robust CPP baseline for determination of U0126 effects. Thus, the 6 CPA mice were removed from the microinjection study prior to VEH/U0126 infusion. The reviewer appears to be confusing the mice from the immunoblotting experiment (Fig 1) and the mice from the microinjection experiment (Fig.2). These are distinct cohorts of animals and completely separate studies (see comment above). A diagram of the procedural time-line is provided in Figure 2. Hopefully, these time-lines will assist the reviewer in navigating the studies.
Line 519-520: Was water available during this 2 hours?
Reply: No, the “classic” DID procedure does not offer water during the 2-h period. Mice have water at all other times of the day.
References are indicated by numbers otherwise, but in the Discussion by names, are these references included in the Reference list?
Reply: As mentioned above, this was an embarrassing mistake on our part and we have now corrected the named citations.
Reviewer 2 Report
This is a very interesting and original work. As the authors indicate on their conclusions, the downstream effects are not known, perhaps in future work they might consider additional biochemical studies, such as whole proteome profiling (or perhaps including phospho-peptide immunopurification), to obtain a better understanding of the biochemical pathways affected.
Author Response
We whole-heartedly agree with this reviewer that more biomolecular work is required in order to more fully understand the role played by ERK-dependent phosphorylation and mGlu5 function in MA reward/reinforcement. The shortage of AA/AA mice precluded even a gross in vivo microdialysis study of their neurochemical phenotype under MA. We hope that pending application for funding of this research will enable our group to delve more deeply into what we think may be a very interesting and clinically relevant story.
Reviewer 3 Report
In this study, Fultz at al. report that mice conditioned with 2mg/kg metamphetamine (MA) i.p., and showing CPP, have elevated ERK1/2 phosphorylation in the mPFC. CPP scores and pERK levels were shown to be positively correlated and the intra-mPFC microinjection of the MEK inhibitor U0126 abolished MA-CPP. Moreover, the authors examined whether the rewarding and reinforcing properties of MA were altered in mice carrying point mutations in the Grm5 gene, that disrupt ERK-phosphorylation of mGlu5 receptors. They observed that despite these mice exhibit markedly increase MA-CPP across a wide range of MA doses, they had lower levels of MA self-administration compared to control mice under operant conditioning procedures. The authors concluded that their study further strengthened the functional relevance of ERK activation for MA-conditioned reward and implicate ERK hyper-activation within mPFC in driving the positive affective/motivational valence of MA.
The authors further claim that ERK-dependent mGlu5 receptor phosphorylation contributes to the propensity to seek out and take MA.
This interesting study addresses an important and timely issue using well established experimental procedures. Methods are sound and the Results clearly illustrated. However, In my view, this study suffers from a number of problems in experimental design and interpretation of the data. Therefore, my enthusiasm for this work is somewhat diminished for the reasons outlined below:
- The attenuation of CPP scores by the intra-mPFC injection of U0126 cannot be claimed to be dose dependent, at least with the dose range used in this study. In fact, there seems to be no significant difference in the CPP score between 1 and 100 nM.
- Whereas the data presented in Fig 2 clearly demonstrate that inhibiting ERK signalling in the mPFC blocks the expression of MA-CPP, they do not directly show a reduction in the positive motivational valence of MA. Hence, the latter claim is not sufficiently substantiated by the data.
- Why was pERK in mPFC not examined in Grm5TS/TS and Grm5AA/AA upon MA-CPP? One of the conclusions of this study is that the mPFC is not a neural locus involved in driving the MA-CPP phenotype of Grm5AA/AA mice. However, how can the authors exclude that this genetic manipulation does not alter mPFC ERK activation? and, furthermore, conclude that ERK phosphorylation of mGlu5 receptors in the mPFC is unrelated to MA-CPP if it has not been directly tested. Indeed, It would have been interesting to investigate, by immunoprecipitation and immunoblotting, whether ERK-phosphorylation of mGlu5 receptors is increased after MA-conditioning.
- As stated by the authors themselves, it is difficult to reconcile the robust MA-CPP with the low-reinforcement phenotype exhibited by the Grm5AA/AA mutant mice. The interpretation that these mice may be more sensitive to MA reinforcement is not convincing as the number of active hole pokes is invariably lower compared to the control mice.
Minor points:
- The murine gene symbol for the metabotropic glutamate 5 receptor is Grm5 and not GRM5 (this symbol is for the human gene). Please change throughout the manuscript.
- To report a concentration of 0 nM (e.g. line 119, 121 etc.) does not make any sense; in such case “vehicle” is the correct expression.
- The number of inactive hole pokes should be shown.
Author Response
- The attenuation of CPP scores by the intra-mPFC injection of U0126 cannot be claimed to be dose dependent, at least with the dose range used in this study. In fact, there seems to be no significant difference in the CPP score between 1 and 100 nM.
Reply: We appreciate this reviewer’s point. Unfortunately, we did not assay a dose between 0 and 1 nM U0126, which would help alleviate their concern. Unfortunately, due to COVID-19 restrictions, we are unable to conduct any follow-up studies at the present time. To satisfy this reviewer, we have removed all mention of dose-dependency when referring to the effects of U0126 on place-conditioning. Please see lines 119 and 328 as examples.
- Whereas the data presented in Fig 2 clearly demonstrate that inhibiting ERK signalling in the mPFC blocks the expression of MA-CPP, they do not directly show a reduction in the positive motivational valence of MA. Hence, the latter claim is not sufficiently substantiated by the data.
Reply: We understand this reviewer’s issue with the term “motivational” when interpreting place-conditioning data. As such, we have removed that term when discussing place-conditioning. We continue to apply the term affective valence, consistent with modern interpretations of place-conditioning behavior (e.g. Prus AJ, James JR, Rosecrans JA. Conditioned Place Preference. In: Buccafusco JJ, editor. Methods of Behavior Analysis in Neuroscience. 2nd edition. Boca Raton (FL): CRC Press/Taylor & Francis; 2009.)
- Why was pERK in mPFC not examined in Grm5TS/TS and Grm5AA/AA upon MA-CPP? One of the conclusions of this study is that the mPFC is not a neural locus involved in driving the MA-CPP phenotype of Grm5AA/AA mice. However, how can the authors exclude that this genetic manipulation does not alter mPFC ERK activation? and, furthermore, conclude that ERK phosphorylation of mGlu5 receptors in the mPFC is unrelated to MA-CPP if it has not been directly tested. Indeed, It would have been interesting to investigate, by immunoprecipitation and immunoblotting, whether ERK-phosphorylation of mGlu5 receptors is increased after MA-conditioning.
Reply: This reviewer raises a number of very important issues and questions for future work.
p-ERK in mutants: No, we did not examine for genotypic differences in ERK expression between the TS/TS and AA/AA mice following place-conditioning. As we hope this reviewer can appreciate, we conducted this study in relatively small cohorts of mice, spaced across a year (size of each cohort varying depending on the size of the shipment and other parallel projects requiring mice). As such, each replicate/cohort of the study only included 1-3 mice/sex/genotype/dose. Thus, it was impossible at the outset of testing to predict the direction of effect of the mutation on the conditioned response to know to sample tissue. Indeed, we have never sampled tissue from the AA/AA mice in our prior cocaine or alcohol work. By the time we completed this behavioral characterization of the AA/AA mice (which followed our characterization of their cocaine and alcohol phenotypes), the Worley laboratory had ceased breeding the animals and subjects were no longer available for any follow-up studies. We know from our published work that both ERK activation and canonical signaling through mGlu5 are intact in the striatum/hippocampus of AA/AA mice (Park et al., 2013), thus we have no reason to believe that the capacity to activate ERK would be altered in our mutants. This issue is complicated further by the fact that, in our hands, repeated methamphetamine alone does not increase ERK phosphorylation in the mPFC (see Neutral and CPA mice in Fig.1; see also Lominac et al., 2016). Thus, without knowing a priori that the mutant mice would exhibit such an augmented MA-CPP, we had no rationale for collecting tissue.
ERK p-mGlu5 and CPP: The reviewer is correct that we cannot conclude 100% that p-mGlu5 is not involved in MA-CPP from the circumstantial data presented in this report. Indeed, we do not make any conclusions in this regard, but rather suggest (based on the fact that the phenotype of the AA/AA mice was polar opposite that produced by an intra-mPFC infusion of U0126) that mGlu5 within the mPFC is not likely the critical site of ERK activity. If mGlu5 was the site, then the phenotype of AA/AA mice would be predicted to at least go in the same direction as the effect of U0126 infusion. Indeed, we observed such parallels in our prior alcohol study of the BNST (Campbell et al., 2019).
p-mGlu5 immunoblotting: We completely agree with the reviewer that it would have been very nice to have a validated phospho-antibody that recognized the amino acid residues in mGlu5 phosphorylated by ERK. We would have loved to examine p-mGlu5 expression in cocaine, alcohol and methamphetamine-treated animals. No such commercial antibody exists. At the time of study (which was 6 years ago), the Worley laboratory had synthesized a p-mGlu5 antibody (see Park et al., 2013) and shipped us some aliquots. However, we could not get the antibody to work in our C57BL/6J tissue on either Bis-Tris or Tris-Acetate gels. Thus, we do not have any data related to p-mGlu5 in drug-experienced animals. If we had research success, those data would have been included in this report.
- As stated by the authors themselves, it is difficult to reconcile the robust MA-CPP with the low-reinforcement phenotype exhibited by the Grm5AA/AA mutant mice. The interpretation that these mice may be more sensitive to MA reinforcement is not convincing as the number of active hole pokes is invariably lower compared to the control mice.
Reply: We appreciate this reviewer’s concern and removed all discussion of the shift to the left in the MA dose-reinforcement curve.
Minor points:
- The murine gene symbol for the metabotropic glutamate 5 receptor is Grm5 and not GRM5 (this symbol is for the human gene). Please change throughout the manuscript.
Reply: We have corrected throughout, including the figure legends. Thank you.
- To report a concentration of 0 nM (e.g. line 119, 121 etc.) does not make any sense; in such case “vehicle” is the correct expression.
Reply: We have replaced 0 nm with vehicle, as suggested.
- The number of inactive hole pokes should be shown.
Reply: We have now included panels in Figure 4 depicting inactive hole-poking. There were absolutely no genotypic differences during any phase of testing.